

# Statistical post-processing of reanalysis wind speeds at hub heights using a diagnostic wind model and neural networks

Sebastian Brune[1] and Jan D. Keller[1,2]

[1]Deutscher Wetterdienst, Offenbach, Germany
[2]Hans-Ertel-Centre for Weather Research, Climate Monitoring and Diagnostics, Germany

**Correspondence:** Sebastian Brune (Sebastian.Brune@dwd.de)

**Abstract.** The correct representation of wind speeds at hub height (e.g., 100m above ground) is becoming more and more important with respect to the expansion of renewable energy. In this study, a post-processing of the wind speed of the regional reanalysis COSMO-REA6 in Central Europe is performed based on a combined physical and statistical approach. The physical basis is provided by downscaling wind speeds with help of a diagnostic wind model, which reduces the horizontal grid point spacing by a factor of eight compared to COSMO-REA6 and considers different vertical atmospheric stabilities.

In the second step, a statistical correction is performed using a neural network as well as a generalized linear model based on different variables of the reanalysis. Although only few measurements by masts or lidars are available at hub height, an improvement of the wind speed in the RMSE of almost 30% can be achieved. A final comparison with radiosonde observations confirms the added value of combining the physical and statistical approach in post-processing the wind speed.

## 1 Introduction

The expansion of wind energy power production is expected to further continue in the context of the ongoing transition towards renewable energies. In order to assess the potential of new sites for wind turbines, reliable estimates of past wind speeds and their variability, i.e., high-quality spatio-temporal climatologies, are needed at hub heights (around 100m above ground, Rohrig et al., 2019). However, deriving a locally meaningful climatology from observations is difficult, as (a) wind speeds have a strong spatial variability and depend on a lot of local characteristics, (b) only few long-term measurements exist in Europe around 100m above ground, and (c) extrapolating hub height wind-speeds from the more abundant 10m wind measurements is prone to errors. In this respect, reanalyses provide physically consistent estimates of the atmospheric dynamics over long periods (i.e., decades). Thus, reanalyses represent a valuable option for assessing wind turbine sites. For this purpose, regional reanalyses might be better suited as they usually use finer horizontal grids which are essential in the description of local effects such as channeling or exposure. Nevertheless, even in such data sets with a horizontal grid spacing of $5 - 10$km, small-scale flows are not always well captured.

Several studies show that some reanalysis data sets have a good fit to verifying mast or lidar observations at hub heights (Frank et al., 2020b; Brune et al., 2021) although larger deviations may occur depending on the location. Further, the underlying





physical models may have systematic errors, e.g., low-level jets is not well represented in the 6km regional reanalysis COSMO-
REA6 (Heppelmann et al., 2017). Therefore, improvements on reanalysis data can be made through statistical post-processing.

Post-processing of wind speed is commonly applied to numerical weather prediction (NWP), but almost exclusively for the
10m wind, which is generally well represented in reanalyses (Kaiser-Weiss et al., 2015). Due to the dense measurement network
for 10m wind speed, local effects as well as synoptic characteristics can be detected and corrected (Jung and Schindler, 2019).
With regard to the wind speed at hub heights of wind turbines, atmospheric stability and turbulent mixing also play an important
role. Brahimi (2019) shows that statistical post-processing of daily wind speeds at hub height using artificial intelligence can
lead to better wind speed estimates.

Another method to improve the horizontal and vertical resolution of wind speed from existing data is to implement a di-
agnostic mass-consistent wind model (Dickerson, 1978; Sherman, 1978; Ratto et al., 1994; Homicz, 2002). The advantage of
this physical approach is that it is able to better describe the effects of orography on the wind field for a given vertical stability
compared to the coarser representation of a NWP model or a reanalysis.

In this study, we combine a diagnostic wind model and statistical post-processing to improve the representation of wind
speeds at 100m above ground despite the low measurement density. Based on the COSMO-REA6 reanalysis (Bollmeyer et al.,
2015) we consider a Central European domain, which includes various different levels of complexity in terrain, e.g., ocean,
flatlands, mid mountain ranges and alpine mountains. Specifically, we aim to answer the following questions:

– Does the introduction of the diagnostic wind model represent an added value?

    – Can we perform a profitable statistical post-processing despite the heterogeneity of the domain and the few measurement
       sites?

The remainder of the paper is structured as follows. In the following section, we first provide an overview of the obser-
vation sites used as well as the COSMO-REA6 regional reanalysis. Then, we describe the wind model and the statistical
post-processing utilizing artificial neural networks in section 3. Our results section begins with an analysis of the effects of
the wind model, followed by the results of the statistical post-processing. We conclude this study with a brief summary and
outlook.

## 2   Data

### 2.1   Mast and Lidar data

Our study is based on a data set of wind profile measurements of the lower boundary layer over Germany and the North
and Baltic Sea. Long-term observations of lower boundary layer wind speeds in Germany are only freely available at four
measuring masts over land and three platforms on the ocean. The land-based masts are located in Hamburg (HAM, Brümmer

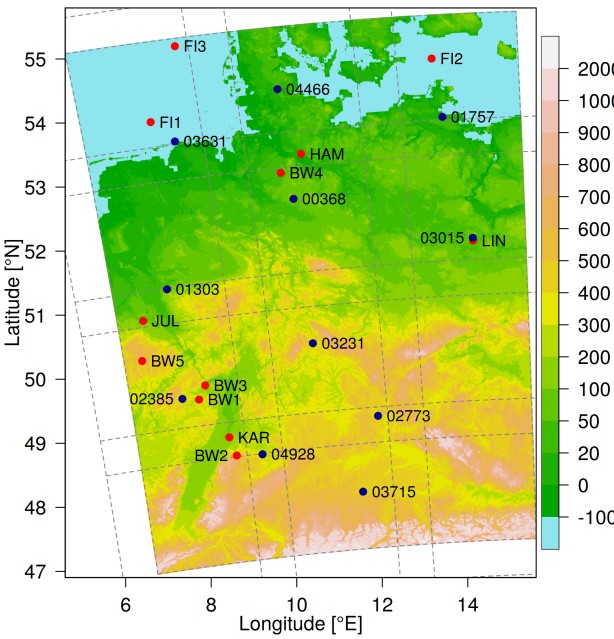

**Figure 1.** Elevation in the study domain (colors) with observation sites (red dots) and radiosondes (blue dots). Dashed lines indicate subdomains of the diagnostic wind model.

et al., 2012)[1], Lindenberg (LIN, Beyrich, 2009)[2], Karlsruhe (KAR, Kohler et al., 2018)[3] and Jülich (JUL, Löhnert et al., 2015; SAMD, 2021)[4] providing data for several decades at heights of up to 280m (Tab. 1). For the North and Baltic Sea, we use the FINO[5] observations (FI1, FI2, FI3) provided by the German Federal Maritime and Hydrographic Agency (BSH, 2021). All three offshore masts capture the complete observation period from 2014 to 2018. The third part of our data set consists of five shorter time series (six to twelve months) performed by Lidars (BW1...BW4) and one meteorological mast (BW5) courtesy of the company BayWa r.e. GmbH. These data are exclusively shared with us within the FAIR project (Frank et al., 2020a).

All measurements are well distributed over the domain (Fig. 1) and represent conditions with offshore (FI1, FI2, FI3), flat terrain (HAM, BW4, LIN) and complex hilly (BW1, BW2, BW3, BW5, KAR, JUL) characteristics. The temporal resolution of all measurements is ten minutes. Additional details on the measurements are provided in Table 1.

---

[1] https://wettermast.uni-hamburg.de/frame.php?doc=Home.htm, last access 22 November 2021

[2] https://www.dwd.de/EN/research/observing_atmosphere/lindenberg_column/boundery_layer/gmfalkenberg_node.html, last access 22 Novemberg 2021

[3] https://www.imk-tro.kit.edu/7791.php, last access 22 November 2021

[4] https://www.fz-juelich.de/gs/DE/UeberUns/Organisation/S-U/Meteorologie/wetter/wstation_node.html, last access 22 November 2021

[5] https://www.fino-offshore.de/en/index.html, last access 08 October 2021



**Table 1.** Overview of mast and Lidar observations.

| Name | Height | Start | End | Type | Environment |
|------|--------|-------|-----|------|-------------|
| BW1 | 98m | 2016-10-10 | 2018-03-20 | Lidar | hilly |
| BW2 | 100m | 2016-10-18 | 2017-10-15 | Lidar | hilly |
| BW3 | 100m | 2018-06-19 | 2018-12-31 | Lidar | hilly |
| BW4 | 102m | 2015-03-03 | 2015-08-04 | Lidar | flat |
| BW5 | 100m | 2015-10-21 | 2016-11-02 | Mast | hilly |
| HAM | 110m | 2014-01-01 | 2015-12-31 | Mast | flat |
| KAR | 100m | 2014-01-01 | 2018-12-31 | Mast | hilly |
| LIN | 98m | 2014-01-01 | 2018-12-31 | Mast | flat |
| JUL | 100m | 2014-01-01 | 2018-12-31 | Mast | hilly |
| FI1 | 102m | 2014-01-01 | 2018-12-31 | Mast | offshore |
| FI2 | 102m | 2014-01-01 | 2018-12-31 | Mast | offshore |
| FI3 | 101m | 2014-01-01 | 2018-12-31 | Mast | offshore |

## 2.2 Radiosondes

Another source of observation data in the height range of wind turbines can be obtained from vertical soundings. The German Meteorological Service (DWD) operates eleven regular radiosondes as shown in Fig. 1 and Table 2[6]. All observations cover the complete period between 2014 and 2018, however, at a much coarser temporal resolution. The radiosondes in Bergen, Idar-Oberstein, Kuemmersbruck and Lindenberg start four times per day at synoptic main times 00:00, 06:00, 12:00 and 18:00 UTC, while observations at other locations arise only twice per day. Note that most radiosondes start approximately 75 minutes before the synoptic main times and that the height of 100 m above surface is already reached after approximately 30 seconds. Thus, we compare the sounding observations with the closest hourly time step of the reanalysis data.

## 2.3 COSMO-REA6

In addition to the observation, our wind speed post-processing relies on gridded estimates of the atmospheric state in the form of the regional reanalysis COSMO-REA6 developed in the context of the Hans Ertel Centre for Weather Research (Simmer et al., 2016). COSMO-REA6 covers Europe at a horizontal grid spacing of 6.2km. The vertical structure is described by a height-based terrain-following coordinate with grid spacing of a few decametres in the lower atmosphere (Bollmeyer et al., 2015). The six lowest levels of 3D data such as temperature, humidity or wind components $u$ and $v$ as well as 2D data are provided through DWD's open data portal (DWD/HErZ, 2021). The hourly output files are available between 01 January 1995 and 31 August 2019.

---

[6]Anzahl der Beobachtungen ergänzen!!!



**Table 2.** Overview of radiosonde observations. The last four columns show the number of observations between 2014 and 2018 at synoptic main times.

| ID | City | Altitude | 04:45 UTC | 10:45 UTC | 16:45 UTC | 22:45 UTC |
|---|---|---|---|---|---|---|
| 00368 | Bergen | 70m | 1661 | 1670 | 1646 | 1644 |
| 01303 | Essen-Bredeney | 150m | - | 1662 | - | 1655 |
| 01757 | Greifswald | 2m | - | 1692 | - | 1664 |
| 02385 | Idar-Oberstein | 376m | 1681 | 1687 | 1689 | 1687 |
| 02773 | Kuemmersbruck | 417m | 1713 | 1709 | 1696 | 1700 |
| 03015 | Lindenberg | 98m | 1661 | 1641 | 1665 | 1663 |
| 03231 | Meiningen | 450m | - | 1671 | - | 1680 |
| 03631 | Norderney | 12m | - | 1616 | - | 1680 |
| 03715 | Oberschleissheim | 484m | - | 1624 | - | 1614 |
| 04466 | Schleswig | 43m | - | 1690 | - | 1450 |
| 04928 | Stuttgart | 314m | - | 1649 | - | 1653 |

Besides both horizontal wind components ($u$ and $v$), we use a set of 16 output variables (Tab. 3) as well as the derived vertical temperature gradient $dT/dz$ within the lowest 100m.

### 2.4 Digital Elevation Data

High-resolution terrain data is freely available through NASA's Shuttle Radar Topography Mission (SRTM). We use the gap-filled version of the SRTM data provided by Jarvis et al. (2008) with a resolution of approximately 90m.

## 3 Methods

### 3.1 Downscaling of COSMO-REA6 wind speed

COSMO-REA6's horizontal resolution with approximately 6km is too low to sufficiently represent orographic effects on the wind field. Therefore, we use a diagnostic mass-consistent wind model which is described in the following.

### 3.1.1 Theoretical background of diagnostic wind modelling

Based on a variational approach (Sasaki, 1958, 1970a, b) the wind model minimizes the variance (kinetic energy) of the difference between the three-dimensional initial wind field $\boldsymbol{v_0}$ and the adjusted wind field $\boldsymbol{v}$ over the volume $V$ as

$$\int_V \frac{1}{2}(\boldsymbol{v} - \boldsymbol{v_0})^2 \rho dV \stackrel{!}{=} \min. \tag{1}$$





**Table 3.** COSMO-REA6 and wind model variables used in the GLMs and NNs with 2, 5, 18 and 21 predictors, respectively.

| Name | Long name | 2 | 5 | 18 | 21 |
|---|---|---|---|---|---|
| ws_COSMO-REA6_0100 | 100m wind speed COSMO-REA6 | x | x | x | x |
| ws_sigw0.0001_0100 | 100m wind speed from wind model (stable atmosphere) | | x | | x |
| ws_sigw0.1000_0100 | 100m wind speed from wind model (neutral atmosphere) | | x | | x |
| ws_sigw5.0000_0100 | 100m wind speed from wind model (unstable atmosphere) | | x | | x |
| dT/dz | Vertical temperature gradient | x | x | x | x |
| CLCH | High cloud cover | | | x | x |
| CLCM | Middle cloud cover | | | x | x |
| CLCL | Low cloud cover | | | x | x |
| CLDEPTH | Vertical extent of clouds | | | x | x |
| RELHUM_2M | 2m relative humidity | | | x | x |
| T_2M | 2m temperature | | | x | x |
| TD_2M | 2m dewpoint temperature | | | x | x |
| VGUST_DYN | 10m maximum wind gusts | | | x | x |
| TWATER | Column integrated water | | | x | x |
| ALB_RAD | Shortwave broadband albedo for diffuse radiation | | | x | x |
| AEVAP_S | Evaporation at surface | | | x | x |
| H_PBL | Height of planetary boundary layer | | | x | x |
| PMSL | Pressure at mean sea level | | | x | x |
| ASOB_S | Net short wave radiation flux at the surface | | | x | x |
| ATHB_S | Net long wave radiation flux at the surface | | | x | x |
| TQV | Vertical integrated water vapour | | | x | x |

The air density $\rho$ is treated as constant in the lower atmosphere and the divergence of the adjusted wind field $v$ should be zero

$$\nabla \cdot v = 0. \tag{2}$$

If we introduce a Langrange multiplier $\lambda = \lambda(x, y, z)$ in Eq.1 under the strong constraint of mass conservation, following cost function $J$ has to be minimized

$$J(u,v,w;\lambda) = \frac{1}{2} \int\limits_V \frac{(u-u_0)^2}{\sigma_u^2} + \frac{(v-v_0)^2}{\sigma_v^2} + \frac{[(w-w_0) - h_x(u-u_0) - h_y(v-v_0)]^2}{\sigma_w^2} dV$$

$$+ \int\limits_V \lambda \left( \frac{\partial u}{\partial x} + \frac{\partial v}{\partial y} + \frac{\partial w}{\partial z} \right) dV \stackrel{!}{=} \min. \tag{3}$$

$u, v, w$ and $u_0, v_0, w_0$ are the components of the three-dimensional adjusted and initial wind field in zonal direction $x$, meridional direction $y$ and vertical direction $z$, respectively. The terms $h_x(u-u_0)$ and $h_y(v-v_0)$ result from the coordinate transformation into a system with a terrain-following vertical coordinate. $h_x$ and $h_y$ are the first derivatives of the topography in $x$ and $y$



direction, respectively. The weights $\sigma_u^{-2}, \sigma_v^{-2}, \sigma_w^{-2}$ are known as Gaussian precision moduli and describe the ratio between the adjustments of the three wind velocity components for the whole domain. Since horizontal wind speeds are generally at least an order of magnitude higher, it is assumed in the literature that $\sigma_u^{-2} = \sigma_v^{-2} \neq \sigma_w^{-2}$ (e.g., Dickerson, 1978; Sherman, 1978; Bhumralkar et al., 1980; Endlich et al., 1982; Guo and Palutikof, 1990; Wang et al., 2005). The ratio $\alpha = \sigma_w / \sigma_u$ determines whether the adjustments are predominantly in the vertical direction ($\alpha >> 1$) or in the horizontal direction ($\alpha << 1$). In an

unstable atmosphere, air motions tend to be vertical, while under stable conditions, adjustments occur predominantly in the horizontal wind field. There are many approaches to determine the exact value of $\alpha$, e.g., using the Froude number (Moussiopoulos et al., 1988; Ross et al., 1988) or determining the ratio of $w$ and $u$ wind (Sherman, 1978; Kitada et al., 1983; Davis et al., 1984; Mathur and Peters, 1990).

     To solve Eq. 3, the first variation of $J$ must be zero. This results in a set of three Euler-Lagrange equations, which can be

written as

$$\boldsymbol{v} - \boldsymbol{v_0} = \mathbf{A}^{-1} \cdot \boldsymbol{\nabla} \lambda \tag{4}$$

with

$$\mathbf{A}^{-1} = \begin{pmatrix} \sigma_u^2 & 0 & h_x \sigma_u^2 \\ 0 & \sigma_v^2 & h_y \sigma_v^2 \\ h_x \sigma_u^2 & h_y \sigma_v^2 & h_x^2 \sigma_u^2 + \sigma_w^2 + h_y^2 \sigma_v^2 \end{pmatrix}. \tag{5}$$

Applying $\boldsymbol{\nabla} \cdot$ on Eq. 4 leads to following Poisson equation for $\lambda$

$$-\boldsymbol{\nabla} \cdot \boldsymbol{v_0} = \boldsymbol{\nabla} \cdot \mathbf{A}^{-1} \cdot \boldsymbol{\nabla} \lambda = \mathbf{M} \lambda. \tag{6}$$

Equation 6 is discretized by using centred differences with lateral flow-through boundary conditions (Diriclet) and no-flow-through boundary conditions at the surface (Neumann conditions). The discretized matrix $\mathbf{M} = \boldsymbol{\nabla} \cdot \mathbf{A}^{-1} \cdot \boldsymbol{\nabla}$ contains only entries on the main diagonal and some subdiagonals, depending on the discrete number of horizontal and vertical grid points. A sparse solver can be used to calculate $\lambda$ and finally the adjusted wind speed $\boldsymbol{v}$ using Eq. 4

$$\boldsymbol{v} = \boldsymbol{v_0} + \mathbf{A}^{-1} \cdot \boldsymbol{\nabla} \lambda. \tag{7}$$

Thus, the main task is to compute $\lambda$ from matrix $\mathbf{M}$, whose dimension is rapidly increasing with the number of horizontal and vertical grid points. Because $\mathbf{M}$ depends only on the Gaussian precision moduli and the topography, the matrix is constant in time and its inverse has to be computed via a sparse factorization once at the beginning. Afterwards the factorized form is used to calculated the adjusted wind field for all time steps.

### 3.1.2   Wind model configuration

As our focus is on Germany and adjacent regions, we first extract a sub-domain of $130 \times 170$ grid points from the COSMO-REA6 data set. The wind model then uses the same domain albeit at a resolution increased by factor of eight, resulting in





a target grid of $1041 \times 1361$ grid points. In the vertical, our wind model uses eleven terrain-following levels (70, 100, 130, 160, 190, 220, 250, 350, 500, 700, and 1000m above the surface). Since the COSMO-REA6 boundary layer winds is strongly influenced by the model orography at the lower two levels (about 10m and 35m above surface), we set the lowest layer in our diagnostic wind model at 70m, which is slightly above the third lowest layer in COSMO-REA6. The COSMO-REA6 wind field is interpolated first vertically and then horizontally to obtain the initial wind field for the wind model.

Consequently, the matrix $\mathbf{M}$ would have a dimension of $15,584,811 \times 15,584,811$ which is too big too handle for the available computing systems. Therefore, we divide the domain into twelve subdomains, each with $401 \times 401 \times 11$ grid points (see Fig. 1), which results in a matrix $\mathbf{M}$ of size $1,768,811 \times 1,768,811$ for each subdomain. The outer 81 points of the subdomains are considered to be the border area. In the transition area between two subdomains, blending of the $u$ and $v$ component is performed, i.e. the influence of the subdomain decreases linearly until the end of the border area. If a border area lies at the edge of the domain, it is truncated so that the final domain has a size of $879 \times 1199 \times 11$ grid points.

To model different degrees of atmospheric stability, we choose $\sigma_u = \sigma_v = 1$ and let $\sigma_w$ vary. After some testing, we settled on three settings, specifically $\sigma_w = 0.0001$ (stable atmosphere, mainly horizontal flow), $\sigma_w = 0.1000$ (relatively neutral atmosphere, similar strong horizontal and vertical flow), and $\sigma_w = 5.0000$ (unstable atmosphere, mainly vertical flow), which is in line with the configuration of Guo and Palutikof (1990).

## 3.2 Statistical modelling using machine learning

While the downscaled wind fields might be better in line with the orography, the data still has inherent uncertainties (e.g., fit of the COSMO-REA6 input to the orography, errors in COSMO-REA6, assumptions in the wind model) and thus may still deviate considerably from the truth, i.e., verifying observations. In order to correct the output of the diagnostic wind model, we apply a simple artificial neural network (ANN) to its output. The ANN consists of an input layer, two dense hidden layers with 50 nodes and a linear activated output layer. For the input and both hidden layers we use the rectified linear activation function. The number of nodes in the input layer varies with the number of input variables. The input variables are scaled in order to set a mean of 0 and a standard deviation of 1 for all parameters. As target variable we choose the deviation between the observed and COSMO-REA6 estimates of wind speed. The error of COSMO-REA6 should be more normally distributed than the wind speed itself which allows us to use the mean squared error as loss function. The optimizer is Adam with a learning rate of 0.001 and a batch size of 256. While we also tried various other configurations for the ANN, e.g., with respect to the number of layers and nodes as well as the different batch sizes, we found the differences in results to be only marginal. Therefore, we here focus on the the ANN settings described above while results for the other configurations are provided in the appendix. For comparison to standard post-processing methods, we also run a generalised linear model (GLM).

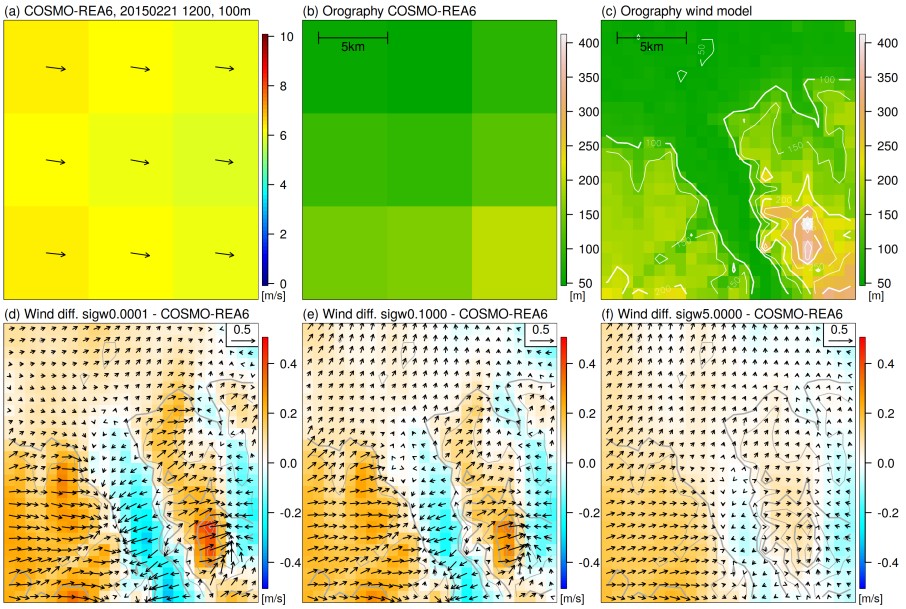

**Figure 2.** Snapshot of COSMO-REA6 wind speed (colors) and direction (arrows) on 21 February 2015 12 UTC in western Germany (a). Both wind components are vertically and horizontally interpolated from the native grid to 100m above surface and the grid box centres. Representation of the topography in COSMO-REA6 (b) and the diagnostic wind model (c). (d)-(f) Difference of the wind field from the diagnostic wind model with $\sigma_w$ values of 0.0001 (d), 0.1000 (e) and 5.0000 (f) to COSMO-REA6. Red (blue) colors indicate a higher (lower) wind speeds in the wind model compared to COSMO-REA6. Arrows show the differences between the wind components in the wind model and COSMO-REA6. The reference vector (top right) represents a difference of $0.5\text{ms}^{-1}$. The grey contour lines represent the topography in the diagnostic wind model.

## 4 Results

### 4.1 Diagnostic wind model

We first look at the potential benefit of applying a diagnostic wind model to the reanalysis output. As an example, Fig. 2 shows the wind representation around the city of Bonn in Western Germany at noon on 21 February 2015 for COSMO-REA6 (a) and the corrections achieved by the wind model for the three different stability settings (d,e,f). The plots show a region of $3 \times 3$ COSMO-REA6 grid points (about 19km $\times$ 19km). The two COSMO-REA6 wind components $u$ and $v$ are first linearly interpolated vertically to 100m above ground and then interpolated from the edges of the grid box to the center. COSMO-REA6 shows uniform wind speeds around $6\text{ms}^{-1}$ from west-northwest directions over the entire region. The underlying orography in 165 the regional reanalysis (Fig. 2(b)) indicates a comparatively flat terrain while the more complex actual terrain structure around Bonn is described by the high-resolution orography of the diagnostic wind model (Fig. 2(c)). In the northern parts and along the Middle Rhine Valley, which extends from southeast to northwest, the elevation is about 50m to 60m above sea level. To the west and east of the valley lie the foothills of the Eifel and Siebengebirge mountains, respectively. The highest elevation in



this region is the Ölberg at 460m which is represented in the wind model with about 410m, while the corresponding pixel in
COSMO-REA6 has only a height of about 200m.

When we interpolate the COSMO-REA6 wind field onto the high-resolution grid and then run the diagnostic wind model,
the differences in horizontal wind speed in this example are up to $\pm 0.5 \mathrm{ms}^{-1}$ at 100m height (Fig. 2(d)-(f)) depending on the
stability setting. This is close to 10% of the COMSO-REA6 wind speed input. The adjustments in the horizontal wind field
are strongest for $\sigma_w = 0.0001$ and decrease with increasing $\sigma_w$. This is consistent with the expectation, since the adjustments
in the wind field for small $\sigma_w$ are almost exclusively horizontal, while for large $\sigma_w$ vertical exchange between model layers is
possible.

The spatial pattern of the wind field is similar for all three configurations of the wind model. In the hilly terrain west and east
of the Rhine Valley we see an increase in wind speeds compared to the reanalysis, while in the valley the wind speed is reduced.
East of the Siebengebirge, i.e. downstream, the wind speed is also lower. In the lowlands, the adjustments are negligible.
Analyzing the wind direction, two interesting features are observed for the stable case ($\sigma_w = 0.0001$). First, there is a flow
around the north and south of the Ölberg, which maybe superimposed by channeling effects in the southeastern part. Second,
the adjustments of the wind field follow the small valley which runs from the lower left corner of the region into the Rhine
valley. Both effects can also be found for the case of the relatively neutral boundary layer ($\sigma_w = 0.1000$), but are absent in the
unstable boundary layer ($\sigma_w = 5.0000$). This indicates that the diagnostic wind model can provide added value in hilly terrain.

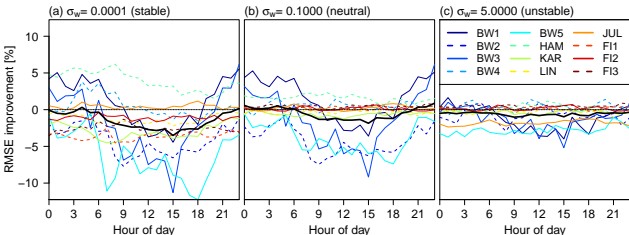

**Figure 3.** Diurnal cycle of the improvement in RMSE of the diagnostic wind model for (a) stable configuration ($\sigma_w = 0.0001$), (b) neutral
configuration ($\sigma_w = 0.1000$ and (c) unstable configuration ($\sigma_w = 5.000$) against COSMO-REA6. Positive (negative) values indicate better
(worse) performance in terms of RMSE of the diagnostic wind model.

Next, we evaluate the quality of the wind field from the diagnostic wind model with measurements. Figure 3 shows the
relative improvement by the wind model with the three configurations for a consistently stable ($\sigma_w = 0.0001$), neutral ($\sigma_w =$
$0.1000$) and unstable ($\sigma_w = 5.0000$) atmosphere against COSMO-REA6. At the offshore observation sites (FI1, FI2, FI3) and
in the lowlands (BW4, HAM, LIN), the wind speeds from the wind model mostly agree with the COSMO-REA6, since only a
few adjustments are made by the model due to the relatively flat terrain. Larger differences in RMSE between COSMO-REA6
and the wind model can be observed in hilly terrain (BW1, BW2, BW3, BW5, KAR, JUL). With higher instability in the
wind model, i.e. increasing $\sigma_w$, the differences in the horizontal wind field are reduced, since the compensating motions are
mainly made in the vertical. Thus, the largest differences between wind model and COSMO-REA6 occur for $\sigma_w = 0.0001$,
where the response of the flow is mainly horizontal. An improvement in RMSE is achieved especially with stable and neutral





configurations between 21 and 06 UTC. This could be an indication that the wind model is able to at least partly correct for the

well-known underestimation of nocturnal low level jets in COMSO-REA6. During the day, COSMO-REA6 exhibits a better performance compared to the diagnostic wind model, especially for the stable and neutral configurations. While COSMO-REA6 performs better than the wind model in about 60% of the cases, improvement can still be found 40% of the time. In order to make use of the additional information, a statistical post-processing is performed using COSMO-REA6 and the outcome of the diagnostic wind model configurations as input.

**4.2 Statistical post-processing of wind speeds at individual locations**

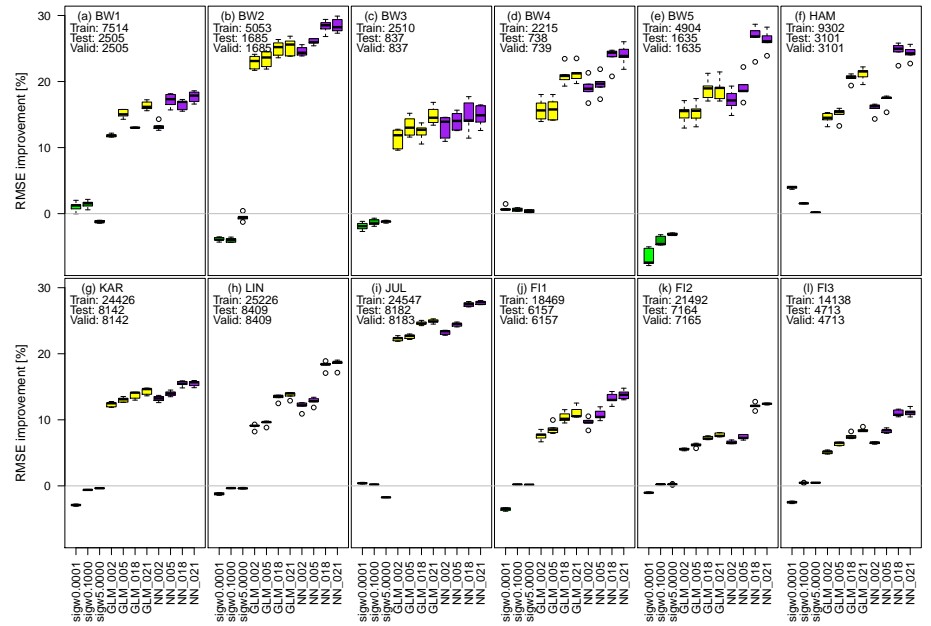

**Figure 4.** The plot shows the change in RMSE compared to COSMO-REA6 for all 12 observation sites with positive values indicating an improvement over the reanalysis. Light green, green and dark green boxplots show the improvement against the COSMO-REA6 from the diagnostic wind model with $\sigma_w = 0.0001$, $\sigma_w = 0.1000$ and $\sigma_w = 5.000$. Yellow and purple boxplots indicate the improvement for the GLMs and NNs, respectively. Each boxplot represent five estimated models obtained by randomly splitting the data set into training, validation and testing. The numbers at the x-axis (2, 5, 18, 21) show the number of input variables for each model. Positive percentages represent an improvement regarding the RMSE against COSMO-REA6. Numbers inside the panels show the sample sizes used for training, testing and validation at each observation site.

Figure 4 shows the enhancement of the post-processing on the RMSE for the diagnostic wind model with the three different stability indices, four GLMs, and four NNs with 2, 5, 18, and 21 input variables at all 12 observation sites. Here, the models are estimated separately for each site. For this purpose, the complete measurement series is randomly divided into 60% training,



20% validation and 20% test. The splitting and estimation of the models is repeated five times to also quantify the uncertainty of the models (indicated with the box plot).

It can be seen that the RMSE for the three diagnostic wind models is close to that of COSMO-REA6. The GLMs and NNs lead to a significant reduction in RMSE at all sites regardless of the number of input variables. For the offshore stations (FI1, FI2, F3) the improvement is at least 5%, while over land the values reach from about 10% in flat terrain (LIN) up to 30% in hilly terrain (BW2). Further, the RMSE reduction becomes more pronounced for the GLMs and NNs as the number of input variables increases with the NNs mostly outperforming the GLMs. It should be noted that the addition of the three wind speed estimates from the diagnostic wind model leads to a significant improvement especially in hilly terrain (e.g., at sites BW1 or BW3) while the effect is smaller at offshore or flat terrain locations (e.g., BW4). Overall, the post-processing, especially with NNs, seems to be capable of achieving a better representation of wind speed compared to COSMO-REA6 regardless of the location.

## 4.3 Statistical post-processing of wind speeds over all locations

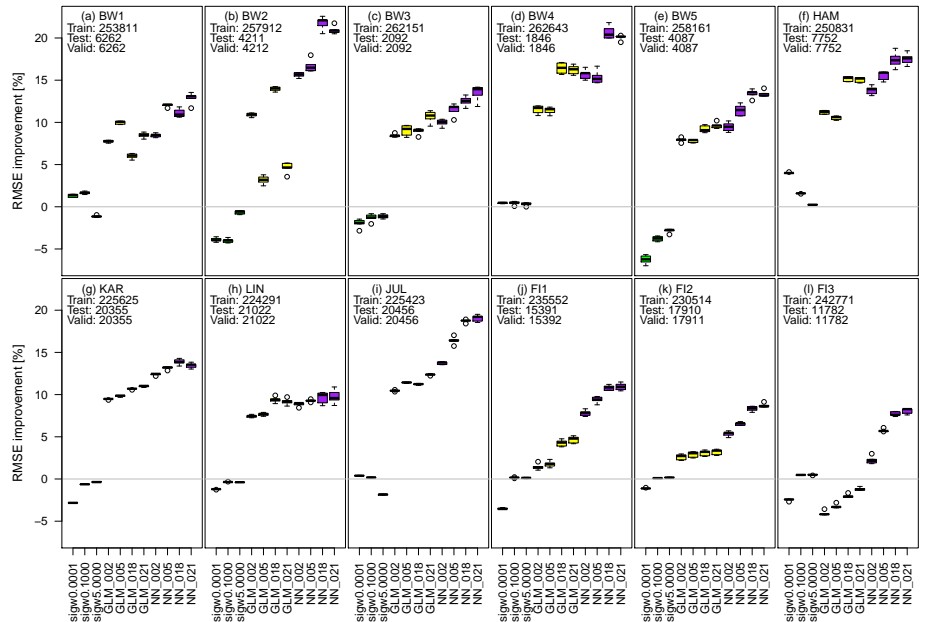

**Figure 5.** As Fig. 4, but now with the training data set of eleven observation sites and the test and validation data set of the site left out.

While the previous post-processing approach is station-specific, it is desirably that such a procedure would be applicable to any random location. Therefore, we now apply a cross-validation approach, i.e., we train the GLMs and NNs on eleven of the twelve locations and use the measurements from the omitted site as validation (50%) and test data set (50%). Thus, the estimated models are evaluated on data from a location not included in the training data.





The effects on the RMSE performance compared to COSMO-REA6 are presented in Fig. 5. Naturally, the improvements are smaller in comparison to the site-specific post-processing as the local characteristics are not included in the cross-validation approach. In this setting, there are now more distinct differences between the performances of the GLMs and NNs. For many stations, the GLM mostly achieves only a small improvement or even leads to a degradation of the quality of the estimates (e.g., FI3). In contrast, the NNs consistently provide better representations of the wind speed compared to COSMO-REA6 as well

as the GLMs. Especially, the NNs with 18 or 21 predictors achieve an improvement of at least 10% (FI1, FI2, FI3) up to about 20% (e.g., BW2, BW4, JUL). The NNs with five predictors are almost always performing better than that with two predictors, indicating the importance of the inclusion of the diagnostic wind model output. However, the 18-predictor version (without the diagnostic wind model data) is outperforming the 21-predictor model at almost half of the observation sites. In conclusion, the diagnostic wind model can add valuable information to the post-processing when only a wind speed and vertical temperature

gradient are used as predictors. However, it seems that the lack of additional information from the diagnostic wind model could be compensated by using a wider set of input variables from COSMO-REA6.

## 4.4 Verification with radiosondes

So far, we have estimated twelve different models by splitting the training and testing data set depending on the observation site. Our final model includes training data from all twelve sites. To prevent the model from being trained primarily on locations

with the most data (due to the different lengths of the time series), the training data covers 2,953 time steps for each location, i.e. 75% of the shortest time series. These data are randomly sampled from the complete time series at each location. In total, we obtain a training (validation) data set with 35,436 (8,844) time steps.

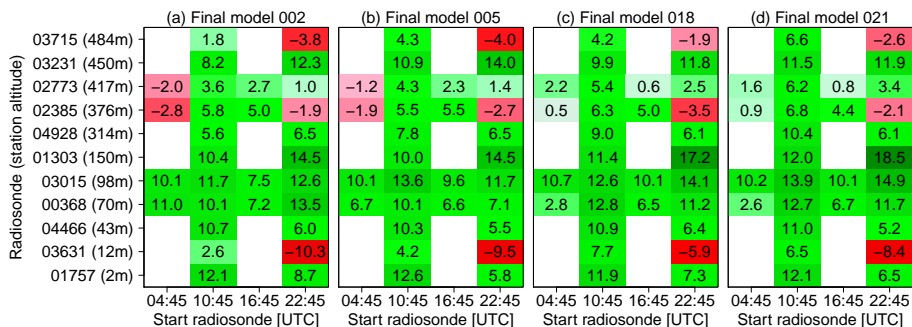

**Figure 6.** Improvement of RMSE in % of the NNs trained over all twelve stations compared to COSMO-REA6 with (a) 2, (b) 5, (c) 18 and (d) 21 input variables, respectively. Green (red) colors indicate an improvement (degradation).

    To evaluate the results, we use observations from radiosondes at eleven sites in Germany. Please note that the radiosonde data has been assimilated into COSMO-REA6 and is only available at certain time steps during the day (c.f. Table 2). Figure 6

shows that the post-processing leads to improvements in terms of RMSE at almost all locations and times, regardless of the number of input variables. While in flat terrain the improvements are smaller, in hilly terrain the skill of the post-processed estimates improves considerably with the number of variables in part due to the added value of the diagnostic wind model. The




model including 21 variables performs particularly well at Essen (01303, almost 20% improvement at night) and Lindenberg
(03015, $10 - 15\%$, depending on the time of day). For the latter, it should be noted that one of the mast locations used to train

the NNs is in proximity to the radiosonde launch site. The NNs seem to have slight difficulties during nighttime for the island
of Norderney (03631, $-8\%$) and in Oberschleissheim (03715, $-3\%$). Both are possibly due to the location of the observation
site directly on the North Sea coast and in the mountains, respectively. Apart from this, the most complex model represents an
improvement of an approximately 8% lower RMSE over all locations and times compared to the COSMO-REA6 reanalysis.
Considering that the radiosonde ascents are already assimilated in COSMO-REA6 and the reanalysis is therefore believed to

perform best at these locations, the results of the post-processing are very encouraging especially with respect to a performance
at locations other then the measurement sites.

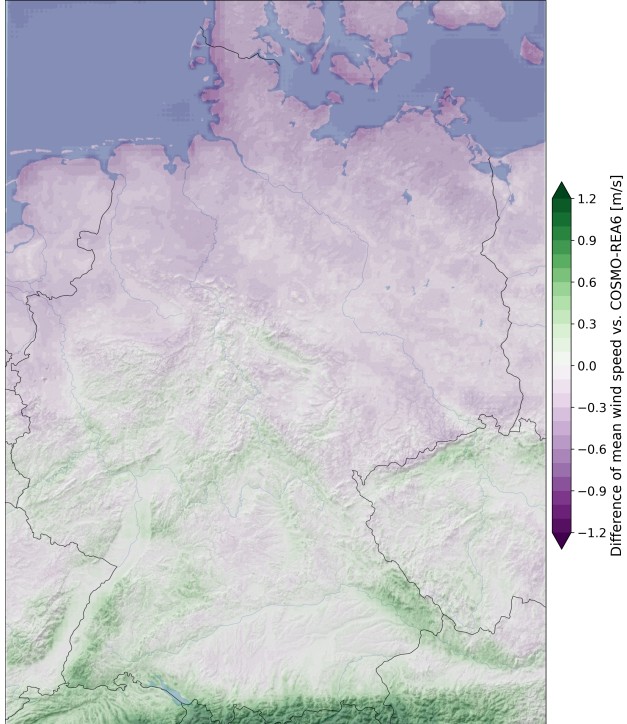

**Figure 7.** Difference of mean wind speed in 2017 between NN_021 and COSMO-REA. Purple (green) colors indicate a decrease (increase)
of post-processed wind speeds compared to COSMO-REA6.

Figure 7 shows the difference of mean wind speed in 2017 for the best post-processing model including 21 variables com-
pared to COSMO-REA6. The corrections by NN_021 result in increased wind speeds over the Alps of more than $1.0 \mathrm{ms}^{-1}$ on
an annual average. The situation is similar for mid-range mountain peaks in Germany, where the corrections are also positive

but somewhat smaller at $0.6 \mathrm{ms}^{-1}$ to $0.9 \mathrm{ms}^{-1}$. This is related to the fact that the small-scale structures of the orography can
be better represented by the considerably higher resolution of the wind model. In the Northern German lowlands, the mean
wind speed is only about $0.3 \mathrm{ms}^{-1}$ below the reanalysis, while the deviations on the North Sea and Baltic Sea coasts are up to





$-1.0 \mathrm{ms}^{-1}$. Since the measurement locations in this study are either offshore (FINO stations) or far inland (all other stations), specific phenomena such as land-sea wind circulation can not be trained by the neural network. Therefore, uncertainties might
be quite large in this area and it may not be possible for the neural network to correctly represent the flow directly along the coast.

## 5 Conclusions

The aim of this study is to enhance the representation of wind speed estimates from reanalysis data around common wind turbine hub heights. By employing a diagnostic wind model to the reanalysis data and using it as additional predictors in a
statistical post-processing approach, we are able to provide a better estimator for wind speed at 100m above ground compared to the COSMO-REA6 regional reanalysis.

We find that the diagnostic wind model alone does not constitute a meaningful improvement on the reanalysis, since it does not take into account the actual stability of the atmosphere but rather corrects wind speeds using three constant vertical atmospheric stability configurations. The added value of the diagnostic wind model only becomes apparent in combination
with the employment of a statistical post-processing approach which combines information from the diagnostic wind model with parameter estimates from the COSMO-REA6 reanalysis (vertical temperature gradient being one of these parameters). We test a generalized linear model as well as different complex neural networks as the statistical modeling framework. In almost all cases, the neural network outperforms the generalized linear model, presumably due to the neural networks ability to include more complex and non-linear interactions between the input parameters.

Further, we have adopted two different types of statistical post-processing models for the wind speed. Specifically, (1) we estimate a separate model for each site, trained on data from the same location only and (2) we train a model on all other eleven sites and then evaluate it at the current site (which is unknown to the model). Both approaches lead to a significant improvement in wind speed estimates. However, the former approach provides better results as local characteristics can only be represented if training data from this location is used. In order to provide estimates at arbitrary locations where no observations are present,
approach (1) is not applicable.

With the encouraging results of the statistical post-processing approach (2), we estimate our final model using data from all twelve observation sites. The estimates are evaluated against radiosonde ascents at eleven locations in Germany. This model yields considerable improvements at most locations (about 8% reduction of RMSE on average), especially when considering that the radiosonde data are already included in the COSMO-REA6 reanalysis. Thus, the combined additional information
from the diagnostic wind model and the statistical post-processing are able to further improve the reanalysis even at locations where COSMO-REA6 is expected to be close to the true state.

As these results are very promising, we now plan to explore the expansion of the current setup to also estimate wind speeds at height levels above 100m. Further, we expect that more improvement might be gained by additional tuning of the statistical model, adding more variables from the reanalysis as predictors, and more observational data including longer time series.





Additional improvement could also be achieved by a more complex diagnostic wind model with more vertical levels and stability parameters.

Nevertheless, our study shows that by combining a physics-motivated approach (i.e., the diagnostic wind model) and a statistical post-processing method (e.g., using artificial intelligence) can be performed at low cost compared to running expensive higher-resolution numerical models. Therefore, the method and derived data sets represent a valuable tool especially for the
wind energy sector, e.g., for yield forecasting or site assessment.

*Data availability.* Selected parameter of the regional reanalysis COSMO-REA6 (DWD/HErZ, 2021) as well as radiosonde data (DWD, 2021) are freely available via DWD's Climate Data Center. Observations of the FINO masts are provided by the German Federal Maritime and Hydrographic Agency (BSH, 2021). Mast observations from Jülich are available within the SAMD archive (SAMD, 2021). Terrain data used in this study is online avaiblable (Jarvis et al., 2008).

**Appendix A: Comparison of NN configurations**

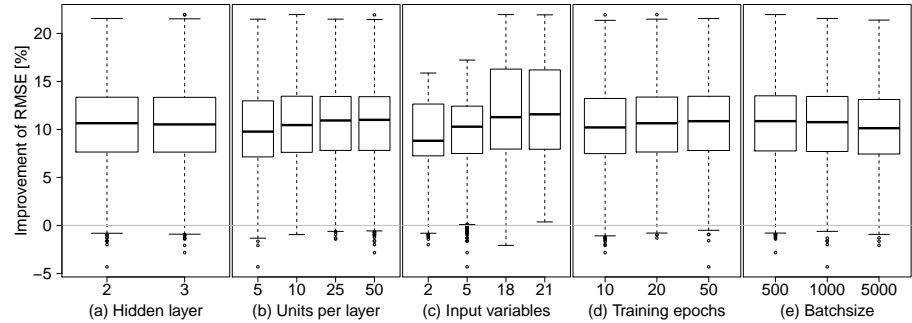

**Figure A1.** RMSE improvements against COSMO-REA6 for all twelve models with five repetitions grouped by (a) number of hidden layer, (b) units per layer, (c) input variables, (d) training epochs and (e) batchsize.

Figure A1 shows the RMSE improvement compared to COSMO-REA6 for all tested configurations grouped by the number of hidden layer, units per hidden layer, number of input variables, training epochs and batchsize for all stations. Increasing the number of hidden layers has no significant effect. The number of units per layer should be 25 or even 50, batchsize 500 or even lower and the number of training epochs should be at least 50. However, the strongest improvement is achieved by adding
more variables, so the exact structure of the neural network is not crucial in the end.

*Author contributions.* SB prepared the data, designed the methodology and carried out the analysis under the supervision of JDK. SB prepared the manuscript. SB and JDK reviewed it iteratively.





*Competing interests.* The authors declare that no competing interests exist.

*Acknowledgements.* This work has been conducted in the framework of the mFund programme funded by the German Federal Ministry for
Transportation and Digital Infrastructure (grant number 19F2103C). The authors want to thank Nicole Ritzhaupt for the support regarding
the diagnostic wind model and BayWa r.e. GmbH (https://www.baywa-re.de/en/) for the generous provision of their data.





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
