# Peer review of "Statistical post-processing of reanalysis wind speeds at hub heights using a diagnostic wind model and neural networks"

_Wind Energy Science, 2022_

## Author Comment (AC1)

We are very thankful both reviewers for their helpful comments and greatly appreciate your suggestions. In the remainder of this response we provide a point-by-point answer (in red) to the reviewers' comments (in black). Again thank you for your suggestions.

**Responses to Reviewer 1 (Michael Mifsud)**

**Comments**

1. Line 48: Could benefit from a literature review, following which the aim of the research is described, rather than being part of the introduction.

   It is our intention to keep the introduction and literature review brief and to summarize the goals of the paper in two concise bullet points within the introduction. Of course, one could think about a more extensive literature review and describe the challenge of wind prediction and wind modeling in more detail. However, from our point of view, the detailed description of the different data (mast/lidar observations, radiosondes, reanalysis, elevation model) and methods (downscaling using the diagnostic wind model, statistical modeling with GLM and ANN) is more relevant for this study. In the description of the methods, we refer to the existing literature and the objective of this study. Therefore, we would like to avoid a long introduction with further subsections in the introduction of this study.

2. lines 88 - 124 could benefit with a description of symbols used.

   We are not sure, what the question aims at. In our view, we have explained all symbols in the text except for $\nabla$ which we now also explained in Eq. (2):

   L95:

   $$\vec{\nabla} \cdot \vec{v} = \left( \frac{\partial}{\partial x}\vec{i}_x + \frac{\partial}{\partial y}\vec{i}_y + \frac{\partial}{\partial z}\vec{i}_z \right) \cdot \vec{v} = 0.$$

   Furthermore, we added the mathematical expressions of $\vec{v}$, $\vec{v}_0$ and $\vec{i}$ in the text:

   L89-90: ... the three-dimensional initial wind field $\mathbf{\tilde{v}_0 = u_0\tilde{i}_x + v_0\tilde{i}_y + w_0\tilde{i}_z}$ and the adjusted wind field $\mathbf{\tilde{v} = u\tilde{i}_x + v\tilde{i}_y + w\tilde{i}_z}$ over ...

   L92-93: $u, v, w$ **and** $u_0, v_0, w_0$ **are the components of the three-dimensional adjusted and initial wind field in zonal direction** $\vec{i}_x$**, meridional direction** $\vec{i}_y$ **and vertical direction** $\vec{i}_z$**, respectively.**

3. Section 3.2: Why ANNs, how do they compare against other Machine Learning models? Why this specific ANN model?

   We use a GLM as a reference for a standard post-processing approach which is itself already a machine learning approach and therefore provides an comparison to another machine learning method. As we want to keep our approach as simple as possible, we use a feed-forward neural network with regular deeply connected layers. It is the most common and frequently

used approach. As can be seen from the manuscript, we explored various configurations of this ANN also detailed in the appendix.

4. Figure 2: Can we see a terrain map near the charts?

   We added the original SRTM terrain map in Fig. 2d for comparison. Unfortunately, we are not allowed to show the exact locations and the surrounding terrain of the measurements used in this study.

5. Figure 3: Assuming RMSE is Root Mean Square Error, how is this being calculated? Why RSME? Why is it better than other metrics such as MAE?

   Correct, RMSE means root-mean-square error. We changed 'RMSE' to 'root-mean-squared error' in the Abstract:

   L7-8: Although only few measurements by masts or lidars are available at hub height, an improvement of the wind speed in the **root-mean-squared error** of almost 30% can be achieved.

   We added a detailed description of the RMSE calculation in the text:

   L186-194: **We employ the standard metric root-mean-squared error (RMSE), which is defined as the sum of squared wind speed difference in the model, i.e. COSMO-REA6 ($c$) or diagnostic wind model ($w$), and the observations ($o$)**

   $$\text{RMSE}_c = \sqrt{\frac{1}{N}\sum_{i=1}^{N}(c_i - o_i)^2}, \qquad \text{RMSE}_w = \sqrt{\frac{1}{N}\sum_{i=1}^{N}(w_i - o_i)^2}. \tag{1}$$

   **$N$ indicates the number of all wind speed measurements. The percentage improvement $\text{PI}_w$ of each wind model $w$ against COSMO-REA6 is then given by**

   $$\text{PI}_w = 100 * \left(1 - \frac{\text{RMSE}_w}{\text{RMSE}_c}\right). \tag{2}$$

   **A smaller RMSE in the wind model compared to COSMO-REA6 leads to $\text{PI}_w > 0$, which indicates an improvement of the diagnostic wind model.**

   The improvement of MAE in percent is very noisy and for certain hours of the day, because the MAE of COSMO-REA6 is close to 0 at some locations. Thus, MAE(model) / MAE(COSMO-REA6) becomes very large (e.g. 4000%). Therefore, we choose the RMSE.

6. What are you comparing in Fig. 3?

   As explained in the text and figure caption, the plots depict the improvement of the diagnostic wind model over the COSMO-REA6 reanalysis data in terms of RMSE. The improvement is now defined by $\text{PI}_{wm}$ and described in more detail (see Comment 5).

7. Figures 4&5: Mainly unclear, you mention colours which are not visible. I have difficulty understanding it.

We agree that the colors are not visible in some boxplots. We decided to fill the background in different colors to differentiate between diagnostic wind model (green), GLM (yellow) and NN (purple).

8. Line 203: Why 60%, 20% 20%? what advantages? is it compared to 70%, 15% 15%?

Interesting point. Indeed, splits of 70-15-15 are more common in literature than 60-20-20. We started our initial experiments with 60-20-20 and were directly happy with our results. After your comment, we tried also a 70-15-15 split. Repeating the model estimation for 20 times with randomly selected train-val-test samples leads to similar results (see following Figure). The median of the 60-20-20 (light grey) and 70-15-15 models (dark grey) are close together, the variance within the 60-20-20 models seems to be slightly smaller. We conclude that the performance within both splits is similar, and the 60-20-20 models may be better generalized. Therefore, we decided to stick with the 60-20-20 split in this study. However, we will consider other splits for future studies.

[Figure]

We added the following sentence in the manuscript:

L213-214: **Our results do not depend on the training-validation-test splitting, as we found in analogous experiments with 70%-15%-15% (not shown).**

9. Assume that NN means neural networks, work could benefit from a list of abbreviations. Difficult to follow all abbreviations.

In the first version of the manuscript we jumped between NN and ANN. We have now used ANN throughout the manuscript to reduce confusion. All abbreviations are now listed at the end of the manuscript (see also following table). Thanks for the suggestion!

Table 1: List of abbreviations.

| Symbol | Long name |
|---|---|
| ANN | Artificial neural network |
| COSMO-REA6 | Regional reanalysis (6km resolution) |
| DWD | German Meteorological Service |
| FAIR | Project to realize a user-friendly exchange of open weather data |
| FINO | Research platforms in the North Sea and Baltic Sea (Forschungsplattformen in Nord- und Ostsee) |
| GLM | Generalised linear model |
| NASA | National Aeronautics and Space Administration |
| NWP | Numerical weather prediction |
| PI | Percentage improvement |
| RMSE | Root-mean-squared error |
| SRTM | Shuttle Radar Topography Mission of NASA |
| UTC | Coordinated Universal Time |

**Responses to Reviewer 2**

**Comments**

Excellent paper, it was a pleasure to read, three minor comments:

1. page 4: delete footnote 6, table 2 is included

   Deleted.

2. page 10 line 173: COSMO instead of COMSO

   Done:

   L.173: This is close to 10% of the **COSMO**-REA6 wind speed input.

3. page 11 line 195: COSMO instead of COMSO

   Done:

   L.194-195: This could be an indication that the wind model is able to at least partly correct for the well-known underestimation of nocturnal low level jets in **COSMO**-REA6.

a review of the revised manuscript is not necessary